# Lack of Spontaneous and Adaptive Resistance Development in *Staphylococcus aureus* Against the Antimicrobial Peptide LTX-109

**DOI:** 10.3390/antibiotics14050492

**Published:** 2025-05-11

**Authors:** Bhupender Singh, Mia Angelique Winkler, Wasifa Kabir, Johanna U Ericson, Arnfinn Sundsfjord

**Affiliations:** 1Research Group for Host-Microbe Interactions, Department of Medical Biology, Faculty of Health Sciences, UiT The Arctic University of Norway, 9019 Tromsø, Norway; mia.angelique.winkler@sus.no (M.A.W.); wasifa.kabir@uit.no (W.K.); johanna.e.sollid@uit.no (J.U.E.); arnfinn.sundsfjord@uit.no (A.S.); 2Centre for New Antibacterial Strategies (CANS), UiT The Arctic University of Norway, 9019 Tromsø, Norway; 3Norwegian National Advisory Unit on Detection of Antimicrobial Resistance, Department of Microbiology and Infection Control, University Hospital of North Norway, 9019 Tromsø, Norway

**Keywords:** antimicrobial agents, antimicrobial resistance, LTX-109, mupirocin, *S. aureus*

## Abstract

Nasal carriage of *Staphylococcus aureus* and its antibiotic-resistant derivative, methicillin-resistant *S. aureus* (MRSA), is a risk factor for nosocomial *S. aureus* infections. Mupirocin is a topical antibiotic and a key in the decolonization of both methicillin-susceptible *S. aureus* (MSSA) and MRSA carriage in patients and health care personnel. Recent observations have shown a global increase in the prevalence of mupirocin-resistant MSSA and MRSA, reducing the efficacy of mupirocin in decolonization regimens. LTX-109 is a peptidomimetic synthetic compound that has shown broad-spectrum bactericidal antimicrobial activity in vitro and in animal experiments. However, the development of resistance against LTX-109 in clinical isolates of MRSA and MSSA has not been systematically examined. Background/Objectives: Here, we assess the development of spontaneous and adaptive resistance against LTX-109 in genomically diverse MRSA (*n* = 3) and MSSA (*n* = 4) strains. Methods: Adaptive and mutational resistance were examined by serial passaging strains over 60 cycles in a range of LTX-109 and mupirocin concentrations. Spontaneous resistance was examined in high-inoculum agar plates with 2–8 times the concentration above MIC. Results: Throughout serial passage, LTX-109 MICs varied less than 4-fold compared to the initial MIC of 4–8 mg/L, while mupirocin MICs increased in all susceptible strains (*n* = 5) from 0.25 mg/L to 16–512 mg/L. The spontaneous resistance assay demonstrated no resistance development at 4–8× MIC LTX-109 and an inoculum effect at 2× MIC. Conclusions: Our results demonstrate the novelty of LTX-109 as an antimicrobial agent with no detectable in vitro resistance development in selected clinical strains of MRSA and MSSA.

## 1. Introduction

*Staphylococcus aureus* (*S. aureus*) is a commensal and an important human opportunistic pathogen causing a wide range of community- and hospital-acquired infections [1]. Multiple skin and mucosal sites of humans can be colonized by *S. aureus*, although the anterior nares are the preferred site of persistent carriage, affecting 20–30% of the adult population [2,3,4]. *S. aureus* nasal carriage is a major risk factor for the development of surgical site (SS) infections, and decolonization strategies have significantly reduced the rate of SS infections caused by *S. aureus* [5,6]. Mupirocin is a widely used antimicrobial in the eradication of methicillin-susceptible *S. aureus* (MSSA) nasal colonization, the prevention of SS infections, and the control of the transmission of methicillin-resistant *S. aureus* (MRSA) in health care institutions [3]. However, a recent meta-analysis has shown a global increase in mupirocin-resistant *S. aureus* (MuRSA) [7]. This includes both an increase in low-level (LLMuRSA) and high-level (HLMuRSA) mupirocin resistance associated with chromosomal mutations in the *ileS* gene and the acquisition of an alternative isoleucyl-tRNA synthetase gene (*mupA* or *mupB*), respectively [8]. The prevalence of MuRSA varied substantially between geographic regions and patient populations, reaching an average above 10% in 12 US studies of clinical strains [7]. Both high- and low-level mupirocin level resistance are associated with *S. aureus* decolonization failure [9]. As the emergence of MuRSA continues to pose challenges for *S. aureus* nasal decolonization, the search for alternative antimicrobial agents has become increasingly important. Among these, LTX-109 is being explored as a potential candidate.

LTX-109 is a chemically synthesized peptidomimetic broad-spectrum drug. Unlike mupirocin, which inhibits RNA and protein synthesis by specifically targeting bacterial isoleucyl-tRNA synthetase, LTX-109 exerts its antimicrobial activity through a membrane-disruptive mechanism [10,11]. It is associated with negatively charged bacterial or fungal membranes, leading to rapid membrane disruption and cell lysis [11]. In vitro activity studies against a large panel of *S. aureus* isolates, resistant to several classes of antimicrobial agents, have shown consistent minimum inhibitory concentrations (MICs) of 2–4 mg/L, irrespective of resistance mechanisms, and rapid concentration-dependent killing [12]. Moreover, a phase I/IIa clinical trial has shown that no safety issues were reported with LTX-109 after 3 days of topical treatment for persistent MRSA/MSSA nasal carriage, supporting the role of LTX-109 as an effective antimicrobial agent in nasal decolonization [13]. It is also effective against bacterial biofilms and bacterial infections in mouse models [14,15]. However, an analysis of resistance development against LTX-109 in *S. aureus* has not yet been systematically examined.

The aim of this study was to investigate the resistance development against LTX-109 in MRSA and MSSA. For this, we (i) conducted broth microdilution (BMD) assays to determine MIC and (ii) examined the occurrence of adaptive and mutational resistance during serial passages as well as (iii) the spontaneous resistance development against LTX-109 and compared it with that of mupirocin.

## 2. Results

The experimental workflow is outlined in Figure 1 and Appendix A and was carefully designed utilizing state-of-the-art methods and recommendations to distinguish various phenotypes [16,17,18,19,20,21,22,23,24].

### 2.1. Broth Microdilution (BMD) Assay MIC Values Were in Agreement with Strain Characteristics

BMD median MIC values are shown in Figure 2 and Appendix A. Median LTX-109 MIC values were 8 mg/L for all the strains, except S5, where it was 4 mg/L, although one of the three S5 biological replicates gave a median MIC of 8 mg/L. All susceptible strains (S1–S5) showed expected mupirocin median MIC values of 0.25 mg/L [25,26]. Mupirocin median MIC values for the strains S6 (LLMuRSA) and S7 (HLMuRSA) were 32 and >512 mg/L, respectively [25,26]. The inoculum size of each strain was within the expected range of around 10^5^ CFU/mL (Appendix A). These BMD assay MIC values served as a baseline for subsequent resistance development assays.

### 2.2. Adaptive and Mutational Resistance in Macrodilution (MAC) Assay Exhibited Steady LTX-109 MIC Values over 60 Passages

MAC assay passage-1 MIC values were comparable to the BMD assay MIC values. For LTX-109, the passage-1 MIC values were 8 mg/L, and over 60 passages, tjeu varied only ±2 fold in strains S1–S7 (Figure 3A and Appendix A).

For mupirocin, the passage-1 MIC values for strains S4 and S5 were 2-fold higher than the BMD assay MIC values. Contrary to LTX-109, mupirocin MAC assay MIC values of strains S1–S5 increased >32-fold within 60 passages, and all MSSA strains acquired a low-level (8–256 mg/L) stable mupirocin resistance phenotype (Figure 3B and Appendix A). However, we observed strain-to-strain variations where all strains, except for S2 and S4, showed a >4-fold increased MIC within 20 passages. After 60 passages, mupirocin MIC in MSSA strains (S1 to S4) and MRSA strain S5 increased to 64, 32, 16, 32, and 64 mg/L, respectively. The *mup*A negative LLMuRSA strain-S6 MIC values also increased 8-fold from 32 to 256 mg/L by passage 14. Notably, the increased mupirocin MICs in mupirocin-evolved strains remained stable during 10 passages performed in the absence of mupirocin, supporting the development of mutational resistance (Figure 4B, Appendix A). Changes in MIC values during 60 passages are shown in Appendix A. In contrast, LTX-109 passaged strains retained MIC values similar to passage-1, ranging between 4 and 8 mg/L (Figure 4A, Appendix A). We randomly selected strains from different passages to confirm inoculum within the range of 10^6^ CFU/mL (Appendix A, Appendix A). The MAC assay results demonstrated a contrasting response of LTX-109 and mupirocin in terms of resistance development, which was further explored in the spontaneous resistance development assay.

### 2.3. Spontaneous Resistance Development Assay (SRD) Indicated a Lack of Resistance Development

All strains grew to form a confluent lawn at 2× LTX-109 MIC following spread. Except for strains S5, S6, and S7, none of the strains showed any growth following streak-02 on 2× MIC of LTX-109. Strains S5, S6, and S7 showed growth following streak-02 in all three, two, and one biological replicates, respectively (Figure 5, Appendix A). The median BMD assay MIC values for all the isolates from streak-02 were similar to their baseline MIC values (Appendix A). None of the strains showed any growth following streak-02 on 4× and 8× MIC of LTX-109. Strain inocula were within the range of 1 × 10^10^ CFU/mL (Appendix A, Appendix A). Since the lack of resistance development in the SRD assay is consistent with the MAC assay results, we, therefore, explored the possibility of tolerance against LTX-109 using the time-kill curve assay.

### 2.4. Time-Kill Curve Assay (TKC) Indicated That the Minimum Duration of Killing 99% Population (MKD_99_) Was <10 Min

We performed TKC assays on selected MSSA (S4) and MRSA (S5) strains; both strains were available at ATCC for use as reference MSSA and MRSA strains. Strain S4 had a median LTX-109 MIC of 8 mg/L in all three replicates (Appendix A). In contrast, strain S5 expressed varying LTX-109 MICs among biological replicates (4 mg/L, 4 mg/L, and 8 mg/L in biological replicates 1, 2, and 3, respectively), with a collective median MIC of 4 mg/L (Appendix A). Since the resolution of the BMD assay is 2-fold, we believe that these variations suggest a true MIC > 4 mg/L for strain S5. Since we used S5 LTX-109 MIC as 4 mg/L, we believed that it might have contributed to the ambiguities that we observed in the SRD assay; therefore, we performed the TKC assay on strain S5 with an extended range of LTX-109 concentrations.

The TKC assay showed strain-to-strain variation at 2× MIC concentrations, with a bacteriostatic effect observed in strain S4 and a bactericidal effect on strain S5. However, we observed a rapid bactericidal activity at 8× MIC in strain S4 and at 4× MIC in strain S5 (Figure 6, Appendix A). In both S4 and S5 strains, there was a 99% reduction in CFUs within 10 min (minimum duration of killing, *MKD*_99_ < 10 min) in all three replicates (Figure 6). At 8× MIC in strain S4 and at 4× MIC in strains S5, LTX-109 showed clear bactericidal activity without any indication of a gradual decrease in bacterial population, which is observed in bacteria exhibiting tolerance [12,16,27,28]. In strain S5, no CFUs were observed at 8× and 16× MIC at the earliest observed time of 10 min (Figure 6, Appendix A). Thus, the TKC assay showed rapid bactericidal action of LTX-109 and a lack of tolerance against it.

## 3. Discussion

Nasal decolonization is important in the prevention of postoperative *S. aureus* infections and the spread of MRSA [3,4,5,6]. Antimicrobial peptides, such as XF-73 and NP-6, are emerging as promising therapeutic agents with affinity against resistant strains of *S. aureus*, and they also have immune-modulating effects to combat bacterial infections [19,29]. Among this growing list of antimicrobial peptides, LTX-109 has shown potential to be an effective topical antimicrobial agent targeting *S. aureus* [13,14,15]. Here, we assessed the ability of LTX-109 against the daunting challenge of combatting resistant *S. aureus* strains by assessing the acquired resistance for 60 serial passages and the development of spontaneous resistance up to 8× MIC in a panel of seven well-characterized MRSA and MSSA strains [30].

To establish a baseline, we first confirmed the mupirocin phenotypes and LTX-109 MIC values using a well-standardized BMD assay. Median mupirocin MIC values of WT and reference strains were 0.25 mg/L (Figure 2, Appendix A). Our observations were in agreement with EUCAST’s WT mupirocin distribution for *S. aureus,* which varies within four 2-fold dilutions (confidence interval 0.125–1 mg/L, median 0.25 mg/L) [26]. MRSA strains S6 and S7 showed median mupirocin MIC values of 32 mg/L and >512 mg/L, reflecting *mup*A independent and dependent mupirocin resistance in S6 and S7, respectively [31]. Interestingly, median LTX-109 MIC values of all strains ranged between 4 and 8 mg/L in BMD and MAC assays (Figure 2, Figure 3 and Appendix A). The LTX-109 MICs were 2-fold higher than previously reported LTX-109 MIC values of 2–4 mg/L [11,12]. MIC values for most WT strains in EUCAST show a distribution, which varies from three to five 2-fold dilutions [17]. Thus, our LTX-109 MICs are well within the range of the upper end of WT MIC or epidemiological cut-off (ECOFF) [17,32,33]. Having established these baseline MIC values, we next examined the potential for resistance development in selected strains.

The most simplistic mechanism for acquired resistance is the selection of chromosomal mutations modifying the antibiotic targets. This is usually examined by testing for the development of adaptive and spontaneous resistance [30]. We did not observe adaptive resistance development against LTX-109 in any strains over a period of 60 passages. The MAC assay LTX-109 MIC values varied only ±2-fold in comparison to passage-1 MIC values for all strains (Figure 3, Appendix A); however, they remained lower than the ECOFF value (Appendix A) [21]. These variations in MIC values in both BMD and MAC assays may be attributed to slight variations in bacterial inoculum (Appendix A). In comparison, mupirocin MIC values increased >32-fold in all susceptible strains and reached an MIC value of ≥16 mg/L, which is of clinical importance [7]. Importantly, the increased mupirocin MIC values by passage 60 remained stable after 10 consecutive passages in the absence of mupirocin (Figure 4, Appendix A), indicating stable acquired mutational resistance.

To complement the MAC assay results, we then investigated the potential for spontaneous resistance development. We did not observe any spontaneous resistant development in any strain at 4× and 8× MIC of LTX-109. At a low MIC (2×); however, all ancestral strains showed a confluent growth (lawn), possibly due to an inoculum effect. The MIC of many antimicrobials has been well documented to be negatively affected by the increase in the number of bacteria (the inoculum size) [34,35,36,37]. In the SRD assay, we used approximately 1 × 10^10^ per mL bacteria as compared to 5 × 10^5^ per mL used to determine the MIC in standard BMD. Thus, the inoculum size was increased by about 1 × 10^5^ times, while the concentration of LTX-109 was increased only 2 times, rendering it relatively ineffective due to a hypothesized threshold number of cell-bound LTX-109 molecules required for bacterial inhibition or killing [37]. Strains S5–S7 grew following streak-02 on media containing 2× MIC of LTX-109 in three, two, and one biological replicates, respectively, in contrast to strains S1–S4 (Figure 5, Appendix A). Notably, from the LTX-109 BMD assay and MAC assay MIC values, strains did show ±2-fold variations (4 or 8 mg/L) within biological and technical replicates, in contrast to strains S1–S4 (8 mg/L in all replicates) (Appendix A). Additionally, 2× MIC values for LTX-109 of these strains were below the calculated ECOFF values [19,21]. Moreover, all isolates from streak-02 showed BMD assay MIC values (Appendix A) similar to their baseline MIC values (Appendix A), which indicated there was no emergence of resistance. Overall, we interpret the growth of strains S5–S7 at 2× MIC as a combined effect of high inoculum and biological variations in BMD assay MIC values, rather than due to the appearance of spontaneous resistance [17,34,35,36]. This notion is further supported by the rapid bactericidal activity of LTX-109 observed at 8× and 4× MIC (Figure 6, Appendix A) in both the reference MSSA (strain S4) and reference MRSA (strain S5), respectively, with similar MKD_99_ values (Figure 6), which do not indicate LTX-109 tolerance in strain S5. The observed dose-dependent response in the TKC assay likely arises from the inherent differences in strain susceptibility towards LTX-109, as also reflected in variations in their baseline MIC values. Thus, both MAC and SRD assays suggest a lack of appearance of any acquired chromosomal mutations against LTX-109 in strains used during the study period. However, we have not analyzed these strains post-LTX-109 exposure by sequencing. Overall, these results demonstrate a lack of resistance development against LTX-109, albeit within the limitations of in vitro monoculture testing.

The strengths in our study include the extensive examination of resistance development against LTX-109 and mupirocin across *S. aureus* strains with differing genetic backgrounds using a robust 60 serial passage MAC assay and an SRD assay. We further examined tolerance development against LTX-109 using state-of-the-art time-kill assays and minimum duration of killing (*MKD*_99_) calculations with negative results. However, in vitro resistance development in monocultures during selective pressure has limitations. In principle, these methods were selected for adaptive and mutational resistance only. Resistance development can also involve horizontal acquisition of new resistance determinants. That needs to be examined in more complex in vitro and in vivo model studies. Further studies should include sequencing-based analyses to adjudicate potential genetic changes following LTX-109 exposure, which might be associated with specific alterations in phenotype. Additionally, in vitro assays do not fully replicate the complexity of in vivo environments. Therefore, careful monitoring of potential resistance development in vivo during clinical investigations is necessary to complement our in vitro observations. Moreover, exploring synergy with other antimicrobial agents could help to assess the potential of combination therapies to reduce the likelihood of resistance development. Finally, clinical validation through well-designed trials is essential to confirm the efficacy and safety of LTX-109, as well as to monitor the potential for resistance development in clinical settings.

## 4. Materials and Methods

### 4.1. Bacterial Strains

In this study, we used seven different strains (designated as strains S1–S7) of *S. aureus,* including three MRSA and four MSSA strains (Table 1). Panels of strains included three wild type (WT; S1–S3) strains isolated during the Tromsø6 *S. aureus* nasal carriage study [38,39,40], one MSSA (S4) and MRSA (S5) reference strain for quality control, and two clinical MRSA (S6–S7) strains with low- and high-level mupirocin resistance, respectively.

### 4.2. Growth Conditions

All strains were grown on horse blood agar (BA) plates and incubated at 37 °C in ambient air for 16–20 h unless specified otherwise.

### 4.3. Antimicrobial Agents

LTX-109 was kindly provided by Pharma Holdings AS (Tromsø, Norway), and mupirocin (catalog No. M7694) was purchased from Merck. The range of dilutions for LTX-109 was 1000 mg/L to 0.125 mg/L and 1024 mg/L to 0.125 mg/L for broth microdilution (BMD) and macrodilution (MAC) assays, respectively. The range of dilutions for mupirocin was 1000 mg/L to 0.015 mg/L and 1024 mg/L to 0.015 mg/L for BMD and MAC assays, respectively. Stock solutions of LTX-109 and mupirocin were prepared in water (750023; ThermoFisher Scientific, Oslo, Norway) and in 10% dimethyl sulfoxide (DMSO) (A13280.36; Alfa Aesar, Oslo, Norway), respectively. We made 2-fold dilutions of antimicrobial agents in Mueller Hinton Broth 2 (MHB; 90922 Merck, Oslo, Norway). Stock antimicrobial solutions and all dilutions were filtered using VWR filtration units (514-0328) fitted with a 0.2 µm filter and were stored at room temperature in the dark until further use.

### 4.4. Broth Microdilution (BMD) Assay

LTX-109 and mupirocin minimum inhibitory concentrations (MICs) were examined by a BMD assay according to the reference method [42,43]. Briefly, BMD assays were performed for each strain in a set of three biological replicates and three technical replicates for each biological replicate using 96-well U-bottomed polypropylene microplates, including negative and positive controls (950261; Greiner BIO-ONE, Oslo, Norway). For each biological replicate, we freshly streaked bacteria from stock culture kept at −70 °C. Bacterial inoculum was prepared by suspending overnight grown bacteria in MHB to obtain a final concentration of 1 × 10^5^/mL to 4 × 10^5^/mL. Working solutions (2× concentrated) of antimicrobials and bacterial suspension were mixed 1:1 in a 100 µL assay volume. Bacterial growth in 96-well plates was assessed manually in accordance with EUCAST guidelines [43]. Bacterial concentration in the inoculum was measured as bacterial colony-forming units per mL (CFU/mL) on blood agar plates following 10-fold serial dilution of bacterial suspension in phosphate-buffered saline (PBS; P4417, Merck) supplemented with 0.05% Triton-X (T8787-Merck; PBS-T).

### 4.5. Adaptive and Mutational Resistance in Serial Macrodilution Assay (MAC)

We examined the development of adaptive resistance against LTX-109 and mupirocin by serial passages in MH-broth macrodilution (MAC) assays. Antimicrobial concentrations during serial passage were gradually increased within a defined range of 0× to 8× MIC to reflect the adaptive resistance development process. BMD assay MIC values were used to select the antimicrobial concentrations for the first passage, ranging from 0× to 8× (0×, 0.5×, 1×, 2×, 4×, and 8× MIC) [18,19,20]. The experimental design is illustrated in Figure 1A. Briefly, bacterial suspensions of 0.5 × 10^6^/mL to 1 × 10^6^/mL cultures were passaged for 60 cycles in a defined range of LTX-109 or mupirocin 2-fold dilutions (0× to 8× MIC) in MHB, or until they acquired a resistance of ≥128 mg/L at 37 °C in ambient air, for 20–22 h, in 1 mL total volume in 12-wells plates (353043, Corning, Oslo, Norway). If the MIC value during a passage increased, the antibiotic concentration for the next passage was increased/adjusted to meet the defined range of 0× to 8× MIC. This ensured that the gradual increase in antimicrobial concentrations was consistent with the resistance development process. MIC values were recorded at the end of each passage. Resistance development was defined as >4-fold increase in initial MIC value [19,21,44]. The presence of adaptive resistance (transiently increased MIC) or a stably resistant phenotype (mutational) with a >4-fold increased MIC was examined by passaging the selected culture at least 10 times without any selection, followed by monitoring MIC values, as mentioned in the MAC assay.

### 4.6. Spontaneous Resistance Development Assay (SRD)

SRD against LTX-109 was examined using a high bacterial inoculum of 1 × 10^10^ CFU, which was spread on 90 mm plates (named “Spread”) containing MH agar supplemented with LTX-109 [22]. The experimental design is illustrated in Figure 1B. MH agar was supplemented with 2×, 4×, and 8× BMD assay LTX-109 MICs. Bacterial growth was observed following incubation of plates overnight at 37 °C in ambient air. The number of colonies per plate was counted and was re-streaked two times on MH plates (named “Streak-01” and “Streak-02”) supplemented with identical LTX-109 concentrations. In cases where bacterial growth was confluent (i.e., forming a continuous layer of growth across the agar plate), individual colonies could not be distinguished. To perform SRD in such cases of confluent growth, bacterial material from agar plates was randomly selected and streaked onto a fresh agar plate (streak-01). A single colony from streak-01 grown bacteria was then streaked again onto a fresh agar plate (streak-02), following the same procedure as described above (Figure 1B). Both streak-01 and streak-02 were performed on fresh MH plates supplemented with identical LTX-109 concentrations. The SRD assay was performed as three independent biological replicates, each consisting of three technical replicates. Bacterial density of the inoculum was evaluated as described. Isolates showing growth following streak-02 were subjected to MIC evaluation using a BMD assay as described above.

### 4.7. Time-Kill Curve Assay (TKC)

TKC assays were performed on strain S4 with 0×, 0.5×, 2×, and 8× BMD assay LTX-109 MICs, and strain S5 had an extended range of 0×, 0.5×, 1×, 2×, 4×, 8×, and 16× BMD assay LTX-109 MICs. The overnight BA grown culture was harvested in 0.9% saline, and cell density was adjusted to approximately 1 × 10^8^/mL. A total of 50 µL of cell suspension in PBS was inoculated in 9.95 mL MHB, and the suspension was incubated at 37 °C in ambient air for 1 h with gentle shaking (50 rpm). A total of 500 µL of the 1 h incubated suspension was diluted with a 2× concentrated antimicrobial agent (1:1; *v*/*v* dilution), in triplicate, in a 12-well plate (734-2778, VWR, Oslo, Norway). A total of 100 µL of cell suspension was withdrawn at 0, 10, 30, 60, 120, and 300 min [23,24,27]. We determined the CFU/mL of the suspension by 10-fold serial dilutions as described earlier. The assay was performed for selected strains in a set of three biological replicates and three technical replicates for each biological replicate. The TKC assay was used to assess the tolerance to LTX-109 by monitoring the ability of the strain to survive a lethal concentration of LTX-109 over a period of 5 h. Tolerance was determined by measuring the minimum duration for killing 99% of the population (*MKD_99_*). Strains with *MKD_99_* values higher than the reference strain were considered as tolerant [16,27].

### 4.8. Statistical Methods Used

Median values were calculated using Microsoft Excel with the formula = MEDIAN (range), where “range” refers to the selected dataset for which the median was determined. For BMD assay MIC assessment, the dataset represents MIC values from three biological replicates, each consisting of three technical replicates. For datasets where averages were applicable, mean values were calculated and plotted using GraphPad Prism (version 10.4.2). Percent CFU reduction in the TKC assay was calculated using Microsoft Excel (version 2502) with the following formula:Percent CFU Reduction = [(Initial CFU − Final CFU)/Initial CFU] × 100
where “Initial CFU” represents the colony-forming units at the start of the assay and “Final CFU” represents the colony-forming units at the indicated time points.

All calculations, including the raw data, intermediate steps, and results, are provided in the Appendix A.

## 5. Conclusions

The overall results support the notion that LTX-109 is efficient against MRSA and MSSA strains with BMD assay MICs of 4–8 mg/L. State-of-the-art adaptive, mutational, and spontaneous resistance development assays did not reveal any resistance development in either MRSA or MSSA against LTX-109. The time-kill curve assay demonstrated a rapid bactericidal killing at 4× to 8× MIC by LTX-109 against MRSA and MSSA, and LTX-109 works in a concentration-dependent manner. These characteristics are compatible with a potentially useful topical antimicrobial agent against MSSA and MRSA.

## Figures and Tables

**Figure 1 antibiotics-14-00492-f001:**
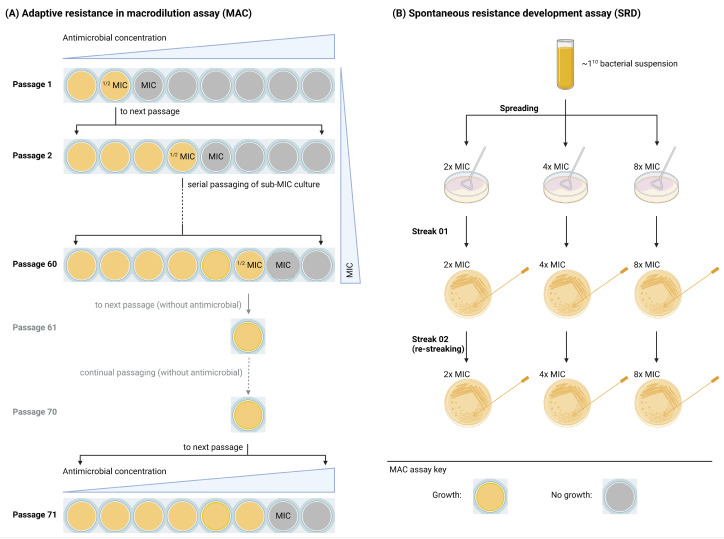
Experimental layout for (**A**) macrodilution (MAC) and (**B**) spontaneous resistant development (SRD) assays. Illustration was created with biorender.com. “https://www.biorender.com/ (accessed on 12 August 2024)”.

**Figure 2 antibiotics-14-00492-f002:**
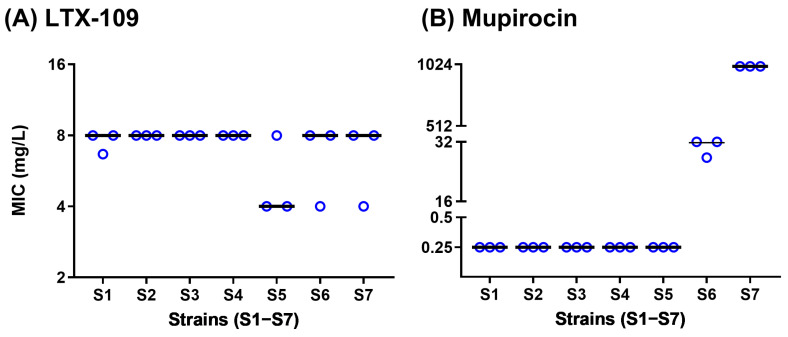
Broth microdilution (BMD) assay MIC values. BMD assay MIC values (mg/L, *Y*-axis) of (**A**) LTX-109 and (**B**) mupirocin for strains S1 to S7 (*X*-axis). The BMD assay was performed in three biological replicates. Each biological replicate is presented as a circle (blue), where the value of the circle is the average of three technical replicates. Median MIC values for each strain from three biological replicates, each consisting of three technical replicates, are represented as solid horizontal lines (black).

**Figure 3 antibiotics-14-00492-f003:**
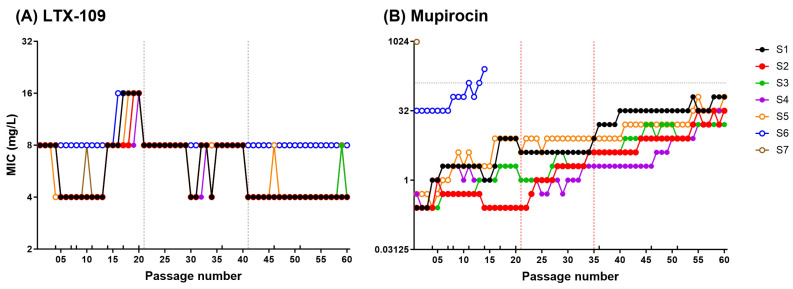
Macrodilution (MAC) assay MIC values over 60 serial passages. MAC assay MIC values (mg/L, *Y*-axis) of LTX-109 (**A**) and mupirocin (**B**) for strains S1–S7 over a period of 60 serial passages (*X*-axis). MIC of each passage is represented as a circle; a dashed horizontal line on the *Y*-axis represents the cut-off MIC value of 128 mg/L. Vertical lines on the *X*-axis represent the use of a new antimicrobial stock solution (dashed black line: LTX-109, and dashed red line: mupirocin). Each data point represents a single measurement of MIC values from strains S1–S7 following each passage, and no statistical analysis was performed.

**Figure 4 antibiotics-14-00492-f004:**
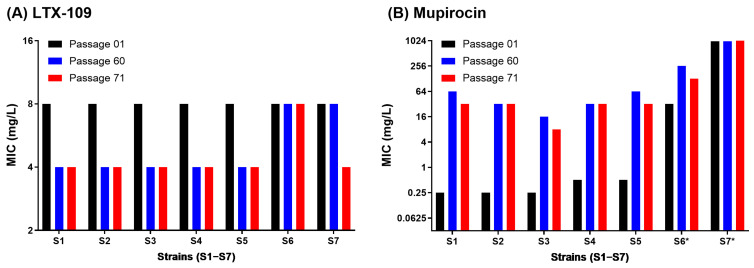
Comparison of MAC assay MIC values before and after 60 serial passages. MIC values at the start of serial passages (passage-1, black bars), after 60 consecutive passages in the presence of LTX-109 (**A**) and mupirocin (**B**) (passage-60, blue bars), and after 10 consecutive passages in the absence of any antibiotic following passage 60 (passage-71, red bars). For mupirocin, passage-60 blue bars for strains S6* and S7* represent MIC values after passages 14 and 1, respectively. MIC values in mg/L are presented on the *Y*-axis for each strain (*X*-axis). Data represent single measurements of MIC values from strains S1–S7 following indicated passages, and no statistical analysis was performed.

**Figure 5 antibiotics-14-00492-f005:**
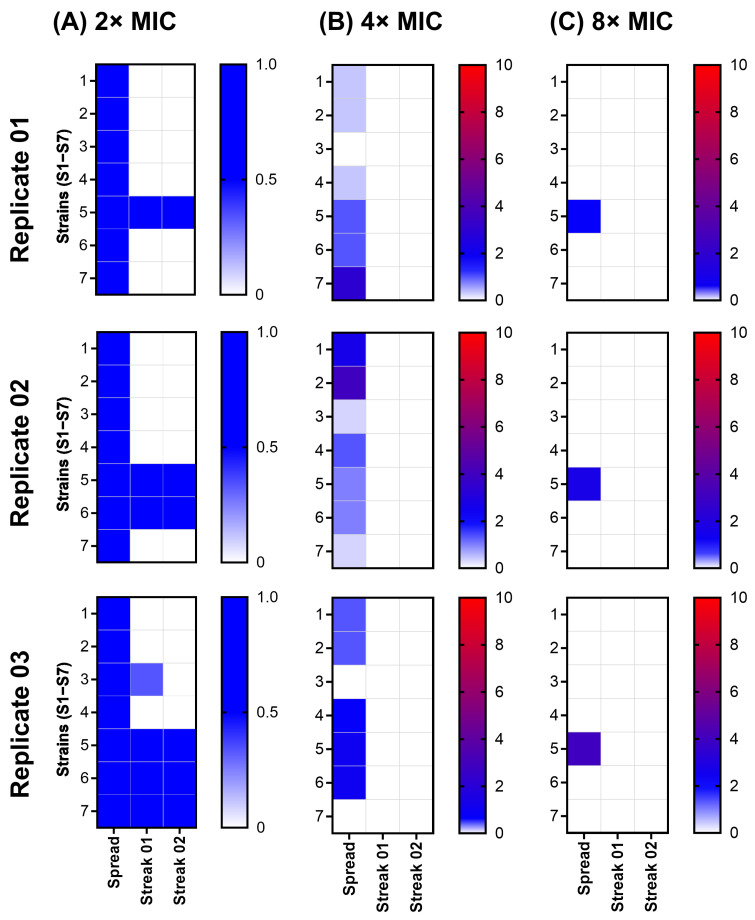
Heat map of average colony count during the spontaneous resistance development (SRD) assay on the MH agar containing 2×, 4×, and 8× MIC of LTX-109 for strains S1 −S7. Each row represents strains S1–S7 (*Y*-axis). The first column of each dataset (2×, 4×, and 8× MIC; *X*-axis) shows either confluent growth (2× MIC) or individual colonies (4× and 8× MIC) following the spread of inocula (spread) on the MH agar. The second column represents the CFU count following streaking (streak-01) of either a random selection from an agar plate if the spread resulted in confluent growth (2× MIC) or all individual colonies (4× and 8× MIC). The third column represents the colony count following a re-streak (streak-02) from streak-01. The scale on the heat map shows either the presence or absence of a confluent layer in three technical replicates (0 = absent, 1 = present) at 2× MIC or the average number of colonies observed from three technical replicates (4× and 8× MIC; scale 0–10).

**Figure 6 antibiotics-14-00492-f006:**
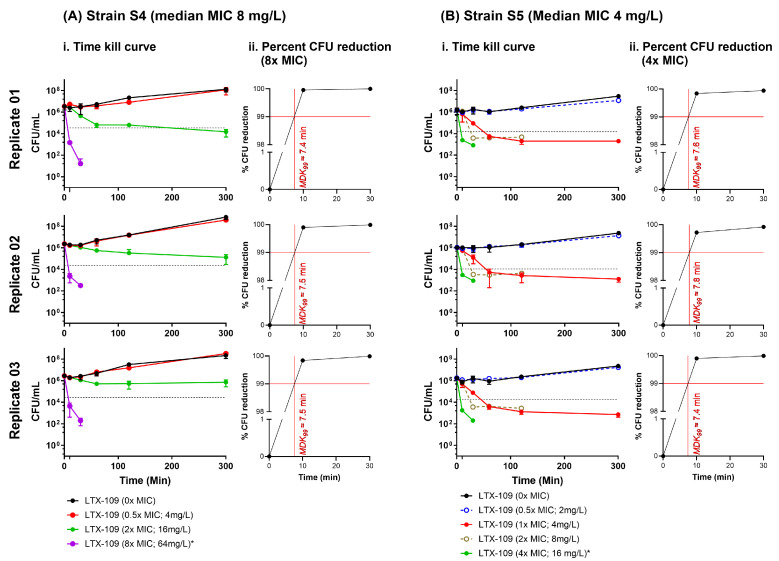
Time-kill curve assay. (**i**) Time-kill curve (TKC) assay for LTX-109 for MRSA strain S4 (**A**) and MSSA strain S5 (**B**). The *Y*-axis represents the average CFU/mL of three technical replicates (±standard deviation) at time points 0, 10, 30, 60, 120, and 300 min (*X*-axis). At 0 min, the bacterial culture of strain S4 was incubated with 0×, 0.5×, 2×, and 8× BMD assay MIC concentrations and of strain S5 with 0×, 0.5×, 1×, 2×, 4×, 8×, and 16× BMD assay MIC concentrations of LTX-109. The black dotted horizontal line indicates a 99% population reduction at 8× MIC; 64 mg/L* in strain S4 and at 4× MIC; 16 mg/L* in strain S5. Note that in strain S5, no CFU was recovered at 10 min for LTX-10 concentrations of 32 mg/L and 64 mg/L. (**ii**) The percentage of CFU reduction was calculated at 8× MIC for MRSA strain S4 (**A**) and at 4× MIC for MSSA strain S5 (**B**). The *Y*-axis represents the percentage of CFU reduction at times points 0, 10, and 30 min (*X*-axis). The red horizontal line indicates a 99% CFU reduction, and the red vertical line indicates the approximation of the time required to achieve 99% CFU reduction (*MKD*_99_).

**Table 1 antibiotics-14-00492-t001:** *Staphylococcus aureus* strains used in this study and relevant characteristics.

Experimental ID	HMI ID ^1^	Spa ^2^	ST ^3^	Sequencing	Origin	Characteristics
S1	68-01	T012	ST30	Pac-bio ^4^	Tromsø6 ^5^	MSSA
S2	68-02	T065	ST45	Pac-bio	Tromsø6	MSSA
S3	68-03	T084	ST15	Pac-bio ^6^	Tromsø6	MSSA
S4	68-04	T021	ST243	Yes ^7^	ATCC 25923	MSSA
S5	68-05	T007	ST39	Yes ^8^	ATCC 43300	MRSA
S6	68-09	T041	N.P. ^11^	Yes, CC5 ^9^	SSI ^10^	MRSA, mupirocin MIC = 16 µg/mL, mupA negative strain
S7	68-11	T067	N.P.	Yes, CC5	SSI	MRSA, mupirocin MIC > 512 µg/mL, mupA positive strain

^1^ HMI ID, bacterial stock identification number at HMI, UiT; ^2^ Spa, *S. aureus* protein A (spa) gene type; ^3^ ST, sequence type pubMLST; ^4^ pac-bio, whole genome sequence (WGS) information available at UiT; ^5^ Tromsø6, stains isolated during the sixth survey of the Tromsø study, 2007-08 [38,39,40]; ^6^ pac-bio, biosample accession number SAMEA112465883, which is associated with the ENA project number PRJEB59355; ^7^ WGS GenBank CP009361 [41]; ^8^ WGS information available from ATCC; ^9^ WGS information available from SSI; ^10^ SSI, Statens Serum Institute, Copenhagen; ^11^ N.P., not provided.

## Data Availability

The raw and supplementary data presented in the study are included in the article Appendix A. Further inquiries can be directed to the corresponding author.

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
