# Peer review of "Lack of Spontaneous and Adaptive Resistance Development in Staphylococcus aureus Against the Antimicrobial Peptide LTX-109"

_antibiotics, 2025, doi:10.3390/antibiotics14050492_

Round 1

Reviewer 1 Report

Comments and Suggestions for Authors

This manuscript is original, scientifically relevant, and addresses an important clinical challenge, the potential for resistance development in Staphylococcus aureus against the synthetic antimicrobial peptide LTX-109. The study is well-organized, methodologically sound, and supported by a comprehensive set of experiments, including MIC determination, adaptive and mutational resistance assessments, spontaneous resistance assays, and time-kill kinetics. The experimental design is rigorous and includes appropriate controls, replicates, and reference strains. The findings are clearly presented and are of potential interest to researchers in antimicrobial drug development, microbiology, and infectious disease prevention. The manuscript also benefits from a clear narrative structure that guides the reader logically through the background, methods, results, and interpretation. Only minor revisions are needed to address some language issues, enhance clarity, and improve the overall flow of certain sections.

General

In introduction, the transition from mupirocin to LTX-109 is slightly abrupt. A smoother bridging sentence would improve coherence.

In methodology, Use of well-characterized clinical and reference strains enhances the generalizability of the results.

In results and discussion, Consistent adherence to MIC standards and resistance interpretation thresholds (e.g., ECOFF values).

In results and discussion, Avoid repetition when discussing MIC variations and assay agreement

In discussion, the mention of the absence of genomic sequencing should be stated earlier, or reiterate its importance when stating final conclusions.

Introduction

38-40: the sentence is overly long and the term "human" appears twice). It could be made more concise for better flow.

60: antimicrobials agents should be “antimicrobial agents”

60-61: a consistent minimum inhibitory concentrations should be “consistent minimum inhibitory concentrations”

63: “no safety issues for” should be should be “with” or “were reported with”.

Material and methods

267: Clearly specify if the stock solutions of LTX-109 and mupirocin were sterile filtered before storage

268: LTX-109 was provided by Pharma Holdings AS, Tromsø. Mupirocin (M7694) was acquired from Merck. I suggest to correct it to be “LTX-109 was kindly provided by Pharma Holdings AS (Tromsø, Norway), and mupirocin (catalog no. M7694) was purchased from Merck.”

287: "Bacterial growth in 96-well plates was measured manually in accordance with EUCAST guidelines." I suggest to replace “measured” with “assessed”

289: "CFU per mL" is used but "CFU/mL" is used also, standardize.

299: Define "ambient air" is it atmospheric COâ‚‚ or not controlled?

314: “In case of confluent growth, we made a random selection from agar plate could be reworded for clarity. Also, try to avoid pronounce such as “we” in scientific writing.

Results and discussion

Logical flow is missing in this section. Add brief transition statements between result subsections and discussion points to improve logical flow.

113: strains remained stably. Should be corrected to “remained stable”

277-278: interpretate the growth. Should be corrected to “interpret the growth”

233: do not support any LTX-109 tolerance in strain S5. I suggest to be corrected to "do not indicate LTX-109 tolerance in strain S5".

Comments on the Quality of English Language

The manuscript is generally well-written and scientifically clear. The English language is of good quality, with appropriate technical terminology and logical sentence structure. However, minor grammatical errors and awkward phrasing are present in several sections and should be corrected during revision to improve overall clarity and readability.

Reviewer 2 Report

Comments and Suggestions for Authors

The manuscript presents a well-structured and comprehensive in vitro study on the resistance development of Staphylococcus aureus strains (both MRSA and MSSA) against LTX-109, a synthetic peptidomimetic antimicrobial. Using a combination of broth microdilution (BMD), macrodilution (MAC), spontaneous resistance development (SRD), and time-kill curve (TKC) assays, the authors demonstrated that LTX-109 did not induce significant resistance in S. aureus over 60 passages, unlike mupirocin which displayed clear adaptive resistance. The results position LTX-109 as a potential alternative for topical use against resistant S. aureus, in particular with regard to nasal application.

Below are some comments and suggestions for improvement:

  1. Introduction: The rationale for comparing LTX-109 with mupirocin is well-justified. However, please provide a clearer explanation of the molecular mechanism of LTX-109 and how it differs mechanistically from mupirocin. In addition, please provide the information regarding the current development state of LTX-109 for application as an antibiotic.
  2. Discussion: To enhance the discussion, please provide a brief comparison with other commonly-applied and emerging antimicrobial peptides. 
  3. Conclusion: While the conclusion is supported by results, it should better reflect limitations (e.g., only in vitro, not sequencing-adjudicated, no in vivo resistance emergence studies) and future research directions (e.g., synergy testing, clinical validation).
  4. Minor revisions: (a) L. 66: “has not yet been thoroughly examined” → “has not yet been systematically examined; (b) L. 208: “remained stable after passaging 10 times” → “remained stable after 10 consecutive passages; (c) ensure consistency in hyphen usage: e.g., "4x-MIC", "8x MIC", "BMD MIC"—standardize for clarity; (d) consider language editing for minor typos and syntax polish.

Thank you.

Reviewer 3 Report

Comments and Suggestions for Authors

The manuscript of Singh et al. presents a detailed explanation of the development of spontaneous and adaptive resistance to antimicrobial peptidomimetic LTX-109 among strains of Staphylococcus aureus. The design of the experiments is well explained and the methods are appropriate for the study objectives. The results are properly explained and well illustrated, and the manuscript content overall is well written.

I recommend publication of this article in Antibiotics after the minor changes detailed below:

1. Please specify in the Methods whether antimicrobial concentrations during serial passage were gradually increased or constant (Section 4.5).
2. Please specify in Sections 4.2 and 4.7 whether bacteria were incubated under ambient air or COâ‚‚ atmosphere so that the methods are completely reproducible.
3. Minor grammatical changes are suggested throughout for better clarity. For example:
• "retained passage-1 like MIC values" → "retained MIC values similar to passage 1"
• "During the serial passaging" → "Throughout serial passage"
4. The captions for the figures (figures 3 and 4) could be improved by including more information to indicate if the data are single measurements or averages, the number of replicates, and if statistical analysis was performed.
5. Check reference formatting for consistency, especially in DOI presentation.
• DOI: In Antibiotics journal format, DOIs should be written as complete hyperlinks beginning with "https://doi.org/'.

Reviewer 4 Report

Comments and Suggestions for Authors

The authors should consider the followings:

The authors should provide the peptide sequence and structure of LTX-109. Also, the rarionales of choosing these peptide over other candidates should be justified.

The authors should provide rationales for the selection of the type and dosing of the antimicrobial agents used in this study.

The authors should justify for those results against the dose-dependent manner. 

Where possible, statistical analayses should be described, i.e. in Figure 2 to 6.

The authors should specify the statistical methodology used in this article.

The authors should revise and minimise the confusion in the list of abbreviation for SSI, where it represented both "Surgical site infections" and " Statens Serum Institute, Copenhagen" in this article.

The authors should provide the method and acceptance criteria for the verification of the strains using in this study, as of those listed in Table 1. "Staphylococcus aureus strains used in this study and relevant characteristics. "

The authors should clarify more about the roles of funder, "Pharma Holdings AS, Tromsø, grant 

number 2021/2543." for this study. Further clarification of the followings is needed "4.3 Antimicrobial agents LTX-109 was provided by Pharma Holdings AS, Tromsø. Mupirocin (M7694) was acquired from Merck. "

The limitations of the study should be clearly listed and discussed.

Novelty of the study should be specified in the abstract.

Comments on the Quality of English Language

Quality of English can be improved.
